# *APC* Mutation Marks an Aggressive Subtype of *BRAF* Mutant Colorectal Cancers

**DOI:** 10.3390/cancers12051171

**Published:** 2020-05-06

**Authors:** Lochlan J. Fennell, Alexandra Kane, Cheng Liu, Diane McKeone, Winnie Fernando, Chang Su, Catherine Bond, Saara Jamieson, Troy Dumenil, Ann-Marie Patch, Stephen H. Kazakoff, John V. Pearson, Nicola Waddell, Barbara Leggett, Vicki L. J. Whitehall

**Affiliations:** 1QIMR Berghofer Medical Research Institute, Queensland 4006, Australia; alexandra.kane@qimr.edu.au (A.K.); john.liu@qimr.edu.au (C.L.); diane.mckeone@qimr.edu.au (D.M.); Winniefernando@gmail.com (W.F.); Chang.su@uqconnect.edu.au (C.S.); Catherine.bond@qimr.edu.au (C.B.); saara@coolbeansunderwear.com (S.J.); troy.dumenil@qimr.edu.au (T.D.); Ann-marie.patch@qimr.edu.au (A.-M.P.); Stephen.kazakoff@qimr.edu.au (S.H.K.); john.pearson@qimr.edu.au (J.V.P.); nic.waddell@qimr.edu.au (N.W.); barbara.leggett@qimr.edu.au (B.L.); Vicki.whitehall@qimr.edu.au (V.L.J.W.); 2School of Medicine, The University of Queensland, Queensland 4072, Australia; 3Conjoint Internal Medicine Laboratory, Chemical Pathology, Pathology Queensland, Herston 4006, Australia; 4Envoi Specialist Pathologists, Queensland 4059, Australia; 5Department of Gastroenterology and Hepatology, The Royal Brisbane and Women’s Hospital, Queensland 4006, Australia

**Keywords:** colorectal cancer, WNT signaling, APC, BRAF, serrated neoplasia, genomics, driver mutations

## Abstract

Background: WNT activation is a hallmark of colorectal cancer. *BRAF* mutation is present in 15% of colorectal cancers, and the role of mutations in WNT signaling regulators in this context is unclear. Here, we evaluate the mutational landscape of WNT signaling regulators in *BRAF* mutant cancers. Methods: we performed exome-sequencing on 24 *BRAF* mutant colorectal cancers and analyzed these data in combination with 175 publicly available *BRAF* mutant colorectal cancer exomes. We assessed the somatic mutational landscape of WNT signaling regulators, and performed hotspot and driver mutation analyses to identify potential drivers of WNT signaling. The effects of *Apc* and *Braf* mutation were modelled, in vivo, using the *Apc^min/+^* and *Braf^V637^/Villin-Cre^ERT2/+^ mouse,* respectively. Results: *RNF43* was the most frequently mutated WNT signaling regulator (41%). Mutations in the beta-catenin destruction complex occurred in 48% of cancers. Hotspot analyses identified potential cancer driver genes in the WNT signaling cascade, including *MEN1*, *GNG12* and *WNT16*. Truncating *APC* mutation was identified in 20.8% of cancers. Truncating *APC* mutation was associated with early age at diagnosis (*p* < 2 × 10^−5^), advanced stage (*p* < 0.01), and poor survival (*p* = 0.026). *Apc^min/+^/Braf^V637^* animals had more numerous and larger SI and colonic lesions (*p* < 0.0001 and *p* < 0.05, respectively), and a markedly reduced survival (median survival: 3.2 months, *p* = 8.8 × 10^−21^), compared to animals with *Apc* or *Braf* mutation alone. Conclusions: the WNT signaling axis is frequently mutated in *BRAF* mutant colorectal cancers. *WNT16* and *MEN1* may be novel drivers of aberrant WNT signaling in colorectal cancer. Co-mutation of *BRAF and APC* generates an extremely aggressive neoplastic phenotype that is associated with poor patient outcome.

## 1. Background

Colorectal cancer is a heterogeneous disease that arises through two main molecular pathways. The conventional pathway, which accounts for 75–80% of all colorectal cancer diagnoses, is initiated by the biallelic inactivation of *APC* and progresses to cancer via mutations in *KRAS* and alterations to the *TP53* gene. By contrast, the serrated neoplasia pathway is initiated by activating mutations in *BRAF* and often progresses to malignancy via *MLH1* hypermethylation, microsatellite instability and a plethora of epigenetic alterations. At the transition to dysplasia, serrated lesions usually acquire mutations that increase WNT signaling. Sessile serrated lesions (SSLs) acquire missense *APC* mutations [1], and truncating *RNF43* mutations [2]. In traditional serrated adenomas (TSAs), common WNT pathway aberrations include *RSPO3* fusions [3,4], mutations of *CTNNB1* [3] and the mutation of *APC* [3].

In the normal enterocytes, the WNT signaling cascade exists to support stemness, differentiation and development. Appropriate levels of WNT signal are maintained intracellularly by the β-catenin destruction complex. The complex consists of AXIN, APC, GSK3*β*, and CK1α. The destruction complex ubiquitinates β-catenin in the cytosol, triggering its subsequent proteasomal degradation. In the absence of the destruction complex, β-catenin translocates to the nucleus and forms a complex with the TCF/LEF molecules and p300 to activate the expression of genes supporting the stem phenotype. Constitutive WNT signaling is deleterious to the cell and thus in the absence of exogenous stimuli the β-catenin destruction complex patrols the cytosol and degrades β-catenin. WNT signaling is activated by the binding of extracellular WNT ligands to frizzled receptors residing on the cell surface. This triggers the sequestering of the destruction complex to the cell membrane and facilitates the build-up of β-catenin, which enters the nucleus and activates WNT target genes.

Approximately 45–50% of *BRAF* mutant cancers demonstrate dysregulated WNT signaling [1], and thus the WNT signaling pathway appears to be important to serrated colorectal neoplasia. In conventional colorectal carcinogenesis, WNT signaling is dysregulated via truncating mutations of *APC* and the loss of 5q21, the region where the *APC* gene resides [5]. This dysregulation occurs very early in the evolution of conventional adenomas. However, numerous studies have indicated that mutation of *BRAF* is almost never identified in such *APC* mutated adenomas, even when they develop advanced histological features [6,7]. This suggests that *BRAF* and *APC* mutations are mutually exclusive in conventional adenomas.

In the serrated neoplastic pathway where the initiating mutation is *BRAF,* WNT signaling only commonly becomes dysregulated when the benign polyp transitions to malignancy. Truncating *RNF43* mutations may alter WNT signaling, but these are predominantly present in mismatch repair deficient *BRAF* mutant cancer [2,8], and there is controversy as to whether *RNF43* mutation affects canonical WNT signaling [9]. The epigenetic silencing of the WNT pathway members is another possible mechanism for altering canonical WNT pathway activity. The methylation of *SFRP* genes increases WNT signaling [10] and is common in colorectal cancer [11]. Similarly, DNA methylation induced the inactivation of *DKK* genes, which are antagonists of WNT signaling, and occur in ~20% of all colon cancers [12]. The frequency of WNT signaling dysregulation being due to *APC* mutation is not well established.

Here we have conducted a large-scale genomic analysis of the somatic mutations that underlie WNT signaling activation in *BRAF* mutant colorectal cancer. We hypothesize that WNT signaling activation in *BRAF* mutant cancers will be heterogeneous, and a mosaic of alterations underpin WNT signaling to achieve a “just-right” level of pathway activation.

## 2. Methods

This study was approved by the QIMR Berghofer Human Research Ethics Committee (P460, P773). Informed consent was obtained for all participants included in this study.

### 2.1. Cohorts Included in the Study

We assessed the somatic mutational landscape of 199 *BRAF* mutant cancers from four distinct sources. This included cancers from the Cancer Genome Atlas project (*n* = 51) [7,13], the Dana Faber Cancer Institute (Giannakis et al. 2016, *n* = 111) [14], the Clinical Proteomic Tumor Analysis Consortium (Suhas et al. 2019, *n* = 13) [15] and additional *BRAF* mutant cancers that were sequenced as part of this study (methods detailed below, *n* = 24). For the survival analyses, we included additional targeted sequenced data from the Memorial Sloan-Kettering Cancer Centre (Yaeger et al. 2018, *n* = 76) [16]. This dataset was limited to a panel of genes, and as such, was excluded from other analyses. Appendix A shows similar tumor mutation burden across each cohort. *BRAF* wild type cancers (*n* = 512) were included for the comparison of mutational profiles between *BRAF* mutant and wild type cancers. Clinicopathological details of samples included in this study and mutational data are available as Appendix A.

### 2.2. DNA Extraction, Library Preparation and Exome Sequencing of Local Samples

Cancer and germline samples were obtained from patients at the Royal Brisbane and Women’s Hospital, Brisbane, Australia, at the time of surgery. All participants gave their written, informed consent prior to participating in the study and the study was approved by the QIMR Berghofer Human Research Ethics Committee (P460, P773). DNA was isolated from whole blood using the salt precipitation method, as previously reported [17]. Cancer samples were snap-frozen in liquid nitrogen and DNA extracted using the AllPrep DNA/RNA/Protein mini kit (QIAGEN, Frankfurt, Germany), as previously reported [18]. Exome-sequencing libraries were generated using the Agilent SureSelect Human All Exon V4+UTR capture platform (Agilent, CA, USA). Libraries were sequenced to a target depth of 200-fold coverage on a 100 bp paired-end sequencing run using an Illumina HiSeq 2000 instrument. Sequence reads were trimmed using Cutadapt (v1.9) [19] and aligned to the GRCh37 reference with BWA-MEM (v0.7.12) [20]. Alignments were duplicate-marked with Picard (v1.129, https://broadinstitute.github.io/picard/) and coordinate-sorted using Samtools (v1.1) [21]. Single nucleotide substitution variants were detected using a dual calling strategy using qSNP (v2.0) [22] and the GATK HaplotypeCaller (v3.3-0) [23]. The HaplotypeCaller was also used to call short indels of ≤50 bp. Initial read filtering for all variants detected included: a minimum of 35 alignment matches in the CIGAR string, 3 or fewer mismatches in the MD field, and a mapping quality greater than 10. High confidence variants were selected with: a minimum coverage of 8 reads in the control data and 12 reads in the tumour data; at least 5 variant supporting reads present where the variant was not within the first or last 5 bases; at least 4 of the 5 reads with unique start positions; the variant was identified in reads of both sequencing directions; the variant was not less than 5 base pairs from a mono-nucleotide run of 7 or more bases in length. Variants were annotated with gene feature information and transcript or protein consequences using SnpEff (v4.0e) [24]. Sequencing and QC metrics are reported in Appendix A.

### 2.3. Assessing the Somatic Mutational Landscape of WNT Regulators

To assess the somatic mutational landscape of WNT signaling regulators, we downloaded mutational annotation files for each cohort from the Genome Data Commons (TCGA), cBioPortal (CPTAC), from Appendix A (DFCI) or analysis of the Royal Brisbane and Women’s Hospital, Brisbane cases. MAF files concatenated to form a combined MAF file comprised of 924,366 entries, relating to 1411 samples. *BRAF* V600E mutant samples were subset from the larger dataset, yielding a total of 320,431 variants from 199 samples. As we sought to investigate WNT, we further selected only genes that were members of the REACTOME signaling by the WNT geneset (*n* = 327 genes). The final dataset was comprised of 5327 nonsynonymous variants in WNT signaling loci that corresponded to 199 samples.

An analysis of variants was performed using the MAFtools R package [25]. Cancer drivers were predicted using two orthogonal approaches (OncodriveClust [26]: default parameters as implemented in MAFtools; OncodriveFML [27]: scores: CADD v1.3, signature: computed by sample, remaining parameters: default). Driver mutation analyses were performed on the entire set of variants (not limited to the WNT signaling gene-set) to accurately model the background mutational processes and avoid statistical biases. Results from Non-WNT loci were discarded and FDR corrections were performed on the remaining *p* values that pertain to tests performed on genes in the WNT signaling pathway. Somatic interactions (co-mutations and mutual exclusivity) were identified by performing Fisher’s exact test on pairs of genes.

### 2.4. Murine Model of Apc and Braf Mutation

To model the effects of *Apc* and *Braf* mutation on colorectal neoplasia, we utilized two murine models. The *Apc*^Min/+^ mouse has a mono-allelic mutation at codon 851 and recapitulates human germline *APC* mutation. In both humans and mice, progression is governed by the loss of the remaining allele. Our second model, the *Braf*^CA/CA^/Villin-Cre^ERT2/+^ mouse [28,29], is an inducible model of *Braf* mutant colorectal neoplasia. Recombination of the mutant *Braf* V637E allele is induced at 2 weeks of age by a single intraperitoneal injection of tamoxifen (75mg/kg). The *Braf* V637E allele is the murine analogue of the *BRAF* V600E human mutation. To model the effects of *Apc* mutation and *Braf* mutation, we crossed *Apc*^Min/+^ mice with *Braf*^CA/CA^/Villin-Cre^ERT2/+^. Animals were monitored biweekly for signs of distress and humanely euthanized when such signs were identified, as per our approved protocol (QIMR Berghofer Animal Ethics Committee; P1208). For survival analysis, animals were deemed to be deceased if they were euthanized due to distress. If animals reached the prescribed endpoints of the experiment without any signs of distress, they were deemed to have survived, and were censored for survival analysis.

At sacrifice, the gastrointestinal tract from oesophagus to rectum was removed, cleaned and opened longitudinally. Macroscopic lesions were bisected to obtain both molecular and histological data. Matched normal hyperplastic tissue was taken minimum five centimetres from the site of the lesion. Histological assessment of lesions and lesion counts was performed on haematoxylin and eosin stained sections from formalin-fixed, paraffin-embedded blocks by specialist gastrointestinal anatomical pathologists.

### 2.5. Sanger Sequencing

Sanger sequencing was performed to assess the G7 repeat track of *WNT16*. PCR conditions were as follows: 1× GoBuffer (ProMega, USA), 2.5 mM MgCl_2_, 0.25 mM dNTP, 0.25 uM Forward Primer (5′ GGCAACATGACAGAGTGTTCC 3′), 0.25 uM Reverse Primer (5′ GCCATACTGGACATCATCGG 3′), 0.25 uM Syto9, 1U GoTaq DNA polymerase (ProMega, USA), 50 ng DNA; Cycing: 95 °C hold for two minutes, 40× cycles of 95 °C for 30 s, 60 °C for 30 s, 72 °C for 45 s, followed by a 72 °C hold for five minutes at the end of cycle 40. Sequencing was performed as per Fennell et al. [30]

### 2.6. Statistical Analysis

All statistical analyses were performed in Microsoft Open R (v3.5.1). Student’s *t* tests were performed for the hypothesis testing of continuous variables. Logistic regression analyses were employed to examine the probability of mutations over patient age. The likelihood–ratio test was performed to assess associations with categorical variables and Fisher’s exact test was used to examine for the mutual exclusivity of mutations in gene-pairs. 

## 3. Results

### 3.1. The Somatic Mutation Landscape of WNT Signaling in BRAF Mutated Cancers

To assess the degree of variation in genetic alterations of WNT signaling pathway genes, we collated whole exome-sequencing data of *BRAF* mutant colorectal cancers from three previously published studies [7,14,15], combined with 24 samples that were sequenced in-house (total *n* = 199). We limited WXS variants to genes in the WNT signaling cascade, as identified in the REACTOME signaling by WNT gene set (*n* = 327 genes). The mean number of WNT pathway mutations per sample was 16.7 ± 13.6, and was highly correlated with overall tumor mutation burden (*p* = 4.08 × 10^−85^, r^2^ = 0.86). *RNF43* was the most commonly mutated gene (41%, Figure 1). Overall, 35.3% of samples had a truncating mutation in *RNF43*. *KMT2D*, *TRRAP,* and *APC* were mutated in 33%, 30% and 28% of samples, respectively (Figure 1).

The *β-catenin* destruction complex is an important regulator of canonical WNT signaling. It is comprised of *APC*, *AXIN1*, *AXIN2*, and GSK3*β* (Figure 2). We next evaluated how frequently mutations occur in any component of this complex or in CTNNB1 itself. Notably, 48% of all cancers had mutations in at least one of these five genes. Mutations in *APC* and *AXIN2*, but not *AXIN1*, are significantly mutually exclusive (*p* < 0.05). Missense mutations in *CTNNB1* have been reported to render the molecule impervious to ubiquitin-mediated destruction. *CTNNB1* mutations occur in 9% of samples. Missense *CTNNB1* mutations were mutually exclusive with truncating *APC* mutations.

*APC* is a classical tumor suppressor gene, where the loss of both copies is required for complete abrogation of transcription. We next assessed the frequency of double, and triple hit mutational events in *APC.* Of cancers with truncating *APC* mutations, 50% also harbored a second truncating mutation in *APC.* One cancer had three truncating mutations in *APC*. When we included missense mutations in the assessment of potential double hit events, this percentage rose to 54%. These data indicate that missense mutations are not selected as second-hits at the *APC* locus, and raise questions regarding their functionality.

### 3.2. Somatic Mutation Interaction Analysis Identifies Co-Mutated WNT Signaling Loci and Mutual Exclusivity of Truncating APC and RNF43 Mutations

We performed somatic mutation interaction analyses to examine for mutations in genes that are mutually exclusive and those that tend to co-occur. We found evidence for co-occurring mutations in 222 gene pairs (Figure 3, Appendix A). As truncating mutations are more likely to influence the final protein, we next examined somatic interactions between truncating mutations in WNT pathway genes (Figure 3). We identified statistical evidence for somatic interactions between 75 gene pairs. 96% were between co-mutated gene pairs (Appendix A). Truncating *APC* mutation was mutually exclusive to truncating mutations in both *RNF43* (P0.0003, OR: 0.20), and ZNRF3 (*p* = 0.001, OR: 0). *AMER1* truncating mutations were mutually exclusive to *RNF43* mutation (*p* = 0.043, OR: 0.12).

### 3.3. The WNT Signaling Mutational Landscape of BRAF Mutant Microsatellite Unstable Cancers Differs from BRAF Mutant Microsatellite Stable Cancers

It is well established that loss of DNA mismatch repair functionality results in a hypermutable phenotype (microsatellite instability, or MSI) and that this is common in BRAF mutant colorectal cancers. We next sought to examine mutations in WNT regulators by microsatellite instability status, to identify if different regulators are targeted in these contexts. We identified fifty genes that were significantly differentially mutated according to microsatellite instability status. As expected, all genes were more frequently mutated in MSI cancers. Table 1 shows the 15 most significantly differentially mutated WNT signaling regulators between MSI and MSS cancers. This analysis reveals genes that are exclusively mutated in MSI cancers (i.e., *TPR2*, *EP300*, *DVL2*) and those that are mutated significantly more frequently in MSI cancers (i.e., *TRRAP*, *RNF43*, *ZNRF3*). The most frequently mutated genes are markedly different between MSI and MSS cancers (Figure 4). The most frequently mutated genes in MSI cancers were *RNF43* (49%), *KMT2D* (48%), and *TRRAP* (41%); in contrast, *APC* was the most frequent mutational target in MSS cancers (21%), followed by *RNF43* (20%), *HECW1* (10%), and a tail of less frequently mutated genes (Figure 4).

### 3.4. WNT Signalling Regulators Are Differentially Mutated in BRAF Mutant Cancers in Comparison to BRAF Wild Type Cancers

To establish whether the WNT mutational landscape of colorectal cancers differs according to *BRAF* status, we examined the mutational profile of WNT signaling genes in 512 further colorectal cancers from the Cancer Genome Atlas that were *BRAF* wild type. The most frequently mutated genes are markedly different in *BRAF* wild type cancers (Figure 1 and Figure 5). *APC* was mutated in 82% of these cancers, in comparison to just 28% of *BRAF* mutant cancers. Of all WNT signaling regulators assessed in this study, 110 were significantly differentially mutated between *BRAF* mutant and wild type cancers (Table 2). Overall, 99% of these genes were more frequently mutated in *BRAF* mutant cancers. This is likely due to the increased frequency of MSI in *BRAF* mutant cancers, and the high frequency of the mutations in highly penetrant *APC* gene in *BRAF* wild type cancers, reducing the selective pressure on other WNT regulators.

### 3.5. Mutation Clustering Analysis Reveals Mutational Hotspots in Nine WNT Signaling Genes

We next sought to identify driver genes using the OncodriveCLUST algorithm. This method identifies potential driver genes using a positional clustering method and operates on the assumption that clusters of mutations, or “mutational hotspots”, are more likely to occur in oncogenes. In keeping with previous studies [2,8,30,31] (p43), *RNF43* was identified as a putative cancer driver (*p* = 0.07). Somatic mutations in *MEN1*, a gene identified as a familial cancer risk gene and as an inducer of genome wide hypermethylation, were identified as putative drivers. *MEN1* was mutated in 4% (8/199, *p* < 0.001) of samples, and most of the identified mutations were frameshift deletions at R521. Moreover, *WNT16* and *GNG12* were implicated as potential cancer drivers (*p* = 0.06 and *p* = 0.0006, respectively).

We used oncodriveFML, an orthogonal computational method of predicting cancer drivers based on predictions of functionality, to identify other potential driver genes that do not necessarily harbor clusters of mutations. This analysis identified 11 potential cancer drivers in the WNT signaling cascade, three of which were identified by oncodriveCLUST (*RNF43*: *p* = 7.22 × 10^−6^, *MEN1*: *p* = 0.02, and *GNG12*: *p* = 0.012). Other genes that were identified include members of the beta-catenin destruction complex (*APC*: *p* = 7.22 × 10^−6^, *AXIN1*: *p* < 0.01, *AXIN2*: *p* = 0.0001), *ZNRF3* (*p* = 7.22 × 10^−6^), *SOX9* (*p* = 7.22 × 10^−6^), *BCL9L* (*p* < 0.001), *PYGO2* (*p* < 0.001), and *WNT11* (*p* = 0.045).

*WNT16* harbors a mutational hotspot at G165. This codon resides in a G7 repeat track that was the subject of frameshift indels in 15 cancers. We used Sanger sequencing to orthogonally validate the presence of *WNT16* hotspot mutations in *BRAF* mutant cancers (*n* = 79) and identified frameshift mutations in 20.2% (16/79) of cancers (Appendix A). Clinicopathological analyses of *WNT16* mutant cancers revealed an association with MSI (mutant: 100% vs. Wt: 61.9%, *p* = 0.0054), right sided tumors (mutant: 100% vs. Wt: 86.8%, *p* = 0.038) and earlier stage at diagnosis (Mutant: 87.6% vs. Wt: 63.2%, *p* = 0.029). When corrected for covariates that may influence cancer stage (MSI, tumour side, sex, age, CIMP), WNT16 mutations were marginally associated with an earlier stage at diagnosis (*p* = 0.05).

### 3.6. Mutations in WNT Signaling Regulators Confer Prognostic Implications

To determine whether frequently mutated WNT regulators, or those identified as putative cancer drivers, is relevant to patient outcome, we performed survival analysis on patients with cancers bearing mutations in the five most frequently mutated WNT signaling regulators (*RNF43, KMT2D, TRRAP, APC, CREBBP)* and those identified by driver analysis (*MEN1, GNG12, WNT16, AXIN1, AXIN2, ZNRF3, SOX9, BCL9L, PYGO2* and *WNT11)*. Four genes (*APC, RNF43, CREBBP and WNT16)* were associated with survival (Table 3). Mutations in *WNT16* and *APC* were significantly associated with shorter overall survival (Table 3). By contrast, mutations of CREBBP and *RNF43* were associated with prolonged survival (Table 3).

### 3.7. Co-Mutation of APC and BRAF Represents a Unique and Aggressive Subtype of BRAF Mutant Cancers

We next evaluated the relationship between *BRAF* mutation and *APC* mutation in further detail, to characterize the clinical and molecular correlates of this subtype of cancers. We supplemented the 199 *BRAF* mutant exomes assessed earlier in the manuscript with 76 *BRAF* mutant cancers that were subjected to targeted sequencing as part of Yaeger et al. 2018 [16]. Truncating mutation was present in 20% of *BRAF* mutant cancers. We examined whether there was a relationship between age at diagnosis and *APC* mutation by logistic regression analysis. The probability of truncating *APC* mutation occurring in a *BRAF* mutant cancer decreases markedly with age from ~60% in patients diagnosed at age 40, to <10% of patients diagnosed at >90 years of age (Logistic Regression *p* = 3.74 × 10^−7^). The average age of patients with a *BRAF^V600E^*/*APC*^Truncated^ cancer was significantly lower than both patients with a *BRAF*
^V600E^/*APC*^Missense^ tumour (61 vs. 72, *p* = 2.03 × 10^−5^, Table 4) and a patient with a *BRAF*
^V600E^/*APC*^Wild-type^ cancer (61 vs. 71, *p* = 9.3 × 10^−6^). *BRAF^V600E^*/*APC*^Truncated^ cancers were more likely to be left sided when compared with *BRAF^V600E^*/*APC^Missense^* cancer (24.5% vs. 4.2%, *p* = 0.02, Table 4). There was no difference in the frequency of CIMP versus either missense or wild-type APC cancers. Overall, 42.3% of *BRAF^V600E^*/*APC*^Truncated^ cancers were microsatellite unstable. MSI is less frequent than both *BRAF*
^V600E^/*APC*^Missense^ (91.3%, *p* = 5.3 × 10^−5^, Table 4) and *BRAF*
^V600E^/*APC*^Wild-type^ (53.8%, *p* = 0.14) cancers.

*BRAF^V600E^*/*APC*^Truncated^ cancers were aggressive cancers, with 67.3% of patients presenting with metastatic disease. In contrast, only 36.4% and 45.7% of *BRAF*
^V600E^/*APC*^Missense^ and *BRAF*
^V600E^/*APC*^Wild-type^ cancers presented at stage III/IV (*p* = 0.01 and 0.002 versus *BRAF^V600E^*/*APC*^Truncated^, respectively). *BRAF^V600E^*/*APC*^Truncated^ cancers that were also microsatellite stable were enriched further for late-stage disease, with 100% of these patients presenting with metastatic disease (Stage III or IV), and 88% with distant metastases (Stage IV).

Furthermore, the univariable analysis of survival indicated that *BRAF^V600E^*/*APC*^Truncated^ cancers have a significantly poorer median survival (504 days vs. 1390 days, log-rank *p* = 0.026, *n* = 32 and *n* = 78 for truncating mutant and wild-type, respectively; Figure 6A). The five-year survival of *BRAF^V600E^*/*APC*^Truncated^ patients was 12%. By contrast, the five year survival of *BRAF^V600E^*/*APC*^Wild-Type^ patients was 42%. This effect was most pronounced in microsatellite stable cancers (Appendix A) compared with microsatellite unstable cancers (Appendix A). We performed a multivariate survival analysis, including age at diagnoses, gender, stage and microsatellite instability as potential prognosticators. Using the Cox-proportional hazard method, microsatellite instability status, and gender are significantly independently associated with survival. Truncating *APC* mutation trends toward conferring independent negative prognostic implications, however this failed to reach the threshold for significance (Table 5, *p* = 0.17). Collectively, these data indicate that the activating mutation of *BRAF* and the truncating mutation of *APC* represent an aggressive subtype of colorectal cancers that occur at a relatively young age in comparison to *BRAF* mutant cancers more generally.

### 3.8. Mutation of Braf in APC^min/+^ Mouse Results in Massive Polyp Load, Rapid Disease Progression and Poor Survival

To determine whether we could recapitulate the apparently aggressive phenotype of the co-mutation of *BRAF* and *APC*, we crossed inducible *Braf^V637^* mutant mice with *Apc^min/+^* mice. The *Braf* mutation was induced at wean in *Apc^Min/+^* mice and we compared the number of lesions per animal and survival to mice with just the mutant *Braf* allele or the mutant *Apc* allele.

We next assessed differences in survival between *Apc^Min/+^ (n = 29), Braf^V63 7^(n = 15), and APC^Min/+^/ Braf^V637^* mice (*n* = 22). Animals were regarded as having survived and were censored if they were healthy at the time of sacrifice; animals were regarded as deceased if the animal had to be euthanized due to illness. Notably, 100% of *Braf* mutant animals survived to 12 months, as did 81.25% of *Apc* mutant animals. Mutation of both *Braf* and *Apc* significantly reduced the survival of the animals (*p* = 8.8 × 10^−21^, Figure 6B). The median survival of animals with both *Apc* and *Braf* mutation was 3.2 months. No animal with both mutations survived longer than six months.

We assessed polyp load by microscopic enumeration. Animals with *Braf* and *Apc* mutations alone develop an average of 4.6 and 16.55 polyps in the small intestine, respectively. Animals with both *Braf* and *Apc* mutation simultaneously develop significantly more lesions in the SI (*p* < 0.0001, Figure 6C). Animals with *Braf* or *Apc* mutation rarely developed colonic or caecal lesions (mean lesions per mouse: 0.11 and 1.1, respectively, Figure 6D). In contrast, the dual mutation of *Apc* and *Braf* resulted in the accumulation of an average of 59.82 colonic/caecal lesions per animal (*p* < 0.0001, Figure 6D). We did not observe a significant increase in lesion size in the small intestine between groups, however we did observe significantly larger lesions in the colon and caecum of animals bearing both *Apc* and *Braf* mutation (*p* < 0.0001, Figure 6E). Lesions had a morphology that was reminiscent of human conventional adenomas, rather than dysplastic serrated lesions.

## 4. Discussion

Here, we have investigated the role of somatic mutation in shaping the WNT signaling landscape of colorectal cancers bearing the *BRAF* mutation. We have shown that 48% of *BRAF* mutant cancers mutate at least one member of the B-catenin destruction complex. Other common modes of activation include the mutation of *RNF43* and *ZNRF3*. We have identified a number of novel mutations that may alter the WNT signaling landscape of cancers. These include *MEN1*, a known WNT pathway tumor suppressor, and *WNT16*, a WNT ligand that may act as an antagonist of ligand mediated WNT activation. Both *MEN1* and *WNT16* harbor hotspot frameshift mutations, that were identified as potential drivers by computational analysis. Mutation of *RNF43* was mutually exclusive to the mutation of *APC*. We examined the clinical and molecular correlates of *BRAF* mutant cancers bearing truncating mutations of *APC,* which occurred in 20% of samples. These cancers were predominantly microsatellite stable, and late stage. Cancers with a truncating *APC* mutation occurred at an average age that was >10 years lower than the wider cohort of *BRAF* mutant cancers. Survival analysis revealed a significantly poorer prognosis for this subtype of patients. Using the murine models of *Apc* and *Braf* mutations, we showed that mutating both genes results in an extensive phenotype with a massive lesion burden. Animals had a median survival of 3.2 months, and no animal bearing both mutations survived longer than six months. Collectively, these data indicate that the mutation of both *BRAF* and *APC* results in an aggressive and rapidly progressing cancer phenotype and confers a poor prognosis.

WNT signaling underpins colorectal carcinogenesis. In the conventional pathway, WNT signaling is usually activated via bi-allelic inactivation of the *APC* tumor suppressor gene at the beginning of the tumorigenic process. However, the mechanisms governing the WNT pathway activation in the serrated neoplasia pathway, which is uniquely marked by *BRAF* mutation, is less clear. In the present study, we sought to identify WNT signaling genes that are mutated in the context of *BRAF* mutant serrated colorectal neoplasia. We obtained exome sequencing data from 175 *BRAF* mutant colorectal cancers from four previously published studies [7,14,15,32] and sequenced a further 24 *BRAF* mutant samples collected locally. Our analyses revealed a mosaic of mutations in WNT signaling regulators, including well-known WNT regulators such as *RNF43, APC, AXIN2 and ZNRF3.* Our analysis identified significant mutual exclusivity between truncating mutations of *RNF43* and *APC.* The mutual exclusivity of mutations in these genes has been previously reported, [8] however it has not been clear whether this mutual exclusivity was, in part, due to the abundance of *RNF43* mutations in *BRAF* mutant cancers, and the relative rarity of *APC* mutations in this context. Here, we have shown that within cancers that bear *BRAF* mutation, the mutually exclusive nature of mutations in these genes remain, indicating that the mutual exclusivity is likely between *APC* and *RNF43* mutations, rather than between *BRAF* mutations and *APC* mutations. Likewise, *ZNRF3* mutation was mutually exclusive to truncating mutations of *APC*. This association was present only when missense mutations were not included. It is possible that the addition of a truncating *APC* mutation in this context is disadvantageous to tumor progression. Therefore, the mutation of *RNF43/ZNRF3* may create a genetic dependency on *APC*. If true, exploiting the dependency on *APC,* a canonical tumor suppressor gene, may be a novel therapeutic treatment for patients with an *RNF43* mutated cancer.

We next examined the exome sequencing data to identify potential novel drivers of WNT signaling activation in colorectal cancer. We adopted a mutational clustering-based approach to identify potential cancer drivers based on the presence of mutational hotspots, as implemented in the OncodriveCLUST algorithm [26]. Reassuringly, *RNF43*, which has two mutational hotspots [2,8,30], was successfully identified as a cancer driver. *RSPO* fusions, which have been implicated in the WNT dysregulation of serrated lesions and cancers [4,33], were not identified due to technological limitations. It is likely that some cancers in this cohort harbored such fusions, given the frequency of RSPO fusions previously reported. We identified eleven other potential cancer driver genes in the WNT signaling cascade. *MEN1* was mutated in eight samples and most mutations were frameshift alterations at codon R521. Germline *MEN1* mutations result in multiple endocrine neoplasia type 1, a tumor predisposition syndrome. It has also been identified as a tumor suppressor gene in a number of different cancer types, including tumors of the parathyroid [34] ^(p1)^, entero-pancreatic neuroendocrine cancers [35] ^(p1)^, and carcinoids [36]. Interestingly, especially in the context of highly methylated *BRAF* mutant cancers, the loss of *MEN1* has been associated with aberrant DNMT1 activity and an altered DNA methylation landscape. To our knowledge, *MEN1* alterations have not been previously reported in colorectal cancer, nor is colorectal cancer a typical presentation of *MEN1* syndrome. It is possible that these patients had an underlying germline mutation in *MEN1,* and the mutations identified in this studied were the second hit at the locus.

*WNT16* was also identified as a potential cancer driver gene. WNT16 is a WNT ligand, a seemingly unlikely candidate tumor suppressor. However, Nalesso et al. [37] showed that while WNT16 was capable of binding Fzd receptors and activating canonical WNT signaling, the degree of activation was significantly lower when compared with the more abundant WNT3A. TOPFlash assays showed that co-stimulation with both WNT3A and WNT16 resulted in significantly less canonical WNT activation when compared with stimulation using WNT3A alone [37]. Thus, it appears WNT16 acts as a competitive inhibitor of Fzd and acts to ensure the homeostasis of WNT signaling. In cancer, the loss of WNT16 may facilitate excessive canonical WNT activation by failure to compete with more potent WNT ligands, such as WNT3A and WNT8. In combination with data indicating poorer prognosis for patients harboring WNT16 mutant cancers, it is likely that WNT16 acts as a tumor suppressor. Inhibitors of WNT ligand secretion, such as porcupine inhibitors, are currently being trialed in colorectal and other solid tumors [38]. Cancers that lack WNT16 are prone to excessive ligand-dependent WNT activation [37] ^(p16)^ and may represent a subset of patients that could benefit from this therapy. Indeed, as much of *WNT16* mutations occur on a background of *RNF43* alterations, which has been shown to confer sensitivity to porcupine inhibitors [31], the mutation of both genes could further sensitize cells to this class of drugs. Although we do not present data in support of this hypothesis in this study, the frequency in which WNT16 is mutated in BRAF mutant colorectal cancers may provide an impetus for investigating whether WNT16 has a role in determining sensitivity to WNT-ligand inhibitors.

We recently assessed a series of 80 *BRAF* mutant cancers [1] and identified truncating *APC* mutation in 11% of these cancers. It has been postulated that truncating APC mutation is uncommon in the context of pre-existing BRAF mutation, because the dysregulation of the WNT signal is too profound in this cellular context. This is consistent with the model proposed by Albuquerque et al. [39] and indicates that mutations such as *RNF43*/*ZNRF3* provide a “just-right” level of WNT signaling to confer a selective advantage. However, the present study has confirmed that a minority of *BRAF* mutant cancers do carry a truncating APC mutation. The higher proportion of cancers bearing both *APC* and *BRAF* mutation in the present study may be due to selection bias in the present series, with a higher proportion of late stage microsatellite stable cancer included.

*APC* mutations were much more common in a minority of *BRAF* mutant cancers diagnosed at a younger age. The average age of *BRAF* mutant cancers harboring *APC* mutation was 12 years less than *APC* wild-type. These cancers were more likely to be microsatellite stable, and present with metastatic disease. The median survival of patients with *BRAF* mutation and *APC* mutation was 64% lower than patients with *BRAF* mutation alone, and patients with both mutations had a five-year survival rate of 12%. *BRAF* mutant microsatellite stable cancers are known to confer a particularly poor prognosis [40], in part due to the low neoantigen burden and poor immunogenicity of the cancers [41]. Although *APC* mutation in *BRAF* mutant cancers was not significantly associated with poorer prognosis on multivariate analysis, we did identify *APC* mutations as prognosticators on univariate analysis. Furthermore, there was a strong and borderline significant trend to a worse still prognosis for *BRAF* mutant MSS cancers with *APC* mutations, compared to those without. Hence, cancers appear to be highly aggressive and occur earlier in life. We generated a murine model to recapitulate mutation of *APC* and *BRAF* to examine interactions between these mutations and the consequences of mutating both genes on polyp development and overall survival. We observed massive polyp loads in animals bearing both mutations, and a markedly reduced survival. Moreover, 100% and 81.25% of *Braf* mutant and *Apc* mutant animals survived to 12 months. When we mutated both genes, no animals survived past six months and the median survival was reduced to a mere 3.2 months. While animals did not develop invasive cancer, and instead died due to polyp load, these data indicate that the commutation of *APC* and *BRAF* in an enterocyte induces rapid neoplastic alterations and an overt proliferative phenotype. Collectively, these data provide strong evidence that the mutation of both *APC* and *BRAF,* whilst uncommon in humans, generates a remarkably aggressive neoplastic phenotype.

It is difficult to resolve how these particular cancers have evolved. Both *APC* and *BRAF* mutations are tumor-initiating events [28,42,43] and give rise to different precursor lesions [44]. Moreover, the cell of origin for *APC* initiated polyps and *BRAF* initiated polyps is hypothesized to differ. *APC* initiated lesions adhere to the “top-down” model [45], whereas *BRAF* mutant lesions are initiated in the stem compartment [46]. Methylation profiling of *BRAF* mutant and *APC* mutant cancers confirmed this model, and showed that *BRAF* mutant cancers had a methylation profile reminiscent of the intestinal stem cell [46] ^(*p*)^. In our study, we observed no difference in the frequency of CIMP between *BRAF* mutants with *APC* mutation and those without, suggesting that these dual-mutant cancers may have arisen in the stem component, and therefore may have been initiated by *BRAF* and acquired an *APC* mutation at a later stage. However, it is difficult to reconcile this with the rarity to which *APC* is mutated in *BRAF* mutant precursor lesions [1]. It is possible that, upon acquiring an *APC* mutation, progression to cancer is rapid and, as a result, identifying lesions in a transitional state is rare. This fits with the aggressive phenotype of these cancers. An alternative hypothesis, supporting an age at diagnosis that is similar to conventional pathway cancers [18] and the morphology of our murine adenomas, is that polyps are initiated by *APC,* acquiring a *BRAF* mutation. If this is the case, such lesions must progress extremely rapidly to cancer, as they are very rarely identified in large series of conventional adenomas.

The mechanisms associated with APC mutation induced aggressiveness in the context of BRAF mutant cancers are not clear. Aberrant WNT signaling appears to be a prerequisite for the acquisition of dysplasia in traditional serrated adenomas [3] and sessile serrated lesion [29,47]. In traditional serrated adenomas, which are by nature dysplastic, are thought to have transitioned from hyperplastic polyps or sessile serrated lesions, and this transition coincides with the acquisition of WNT pathway alterations, such as *RSPO* fusions [4] and *RNF43* mutations [3]. In sessile serrated lesions that acquire dysplasia, *RNF43* mutations are extremely common (>50%) [47], however they are rare in lesions that are not yet dysplastic. This is an important distinction, as lesions with dysplasia are thought to progress within 12 months to invasive cancer [48]. Alterations to RNF43 are dependent on DNA methylation induced loss of MLH1 and mismatch repair function. In sessile serrated lesions, DNA methylation may develop over a protracted period of time, increasing the sojourn of these lesions to dysplasia and ultimately malignancy [49]. *BRAF* mutation has been reported to induce senescence, which can be overcome through silencing of the *CDKN2A* loci [50]. It has been previously reported that *APC* loss in the intestine does require loss of p21 to generate adenomas and does not induce senescence [51]. It is possible that *APC* mutations, in the context of BRAF mutation, facilitate an exit to oncogene induced senescence and while simultaneously satisfying the requirement for WNT signaling, facilitating a rapid transition to dysplasia and invasive cancer. This may explain why precancerous lesions with both *APC* mutation and BRAF mutation are rarely identified, and why the cancers with these mutations present at a younger age.

Our study has comprehensively assessed the somatic mutation landscape of WNT signaling regulators in a large series of *BRAF* mutant colorectal cancers, however several limitations remain. As we assessed publicly accessible sequencing data from various sources, the depth of sequencing, bioinformatic pipeline and variant filtering methods was not uniform across cohorts. This may result in the under (or over) reporting of certain variants. Our study was also limited in the ability to assess the role of the CpG island methylator phenotype in WNT signaling. CIMP was assessed in some samples in our study cohorts, however the method of assessment was not uniform across samples. For example, cancers from TCGA assessed CIMP via microarray based clustering; in contrast, other studies used qPCR based marker panels. Moreover, many WNT signaling genes are putative methylation targets. Further study is necessary to assess the role of CIMP in shaping WNT signaling in the context of *BRAF* mutant colorectal cancer. Although our study is the largest to date to assess WNT regulators in BRAF mutant cancers, we did not have a sufficiently large enough sample size to draw conclusions as to the prognostic implications of genes that are mutated more rarely, such as *MEN1* and GNG12. Mutations in hotspots, or in relatively small genes, could be assessed in much larger cohorts using less expensive genotyping technologies, to determine whether mutations in these genes are relevant to patient outcome.

## 5. Conclusions

In conclusion, here we have conducted a comprehensive survey of the somatic mutational landscape shaping WNT signaling in *BRAF* mutant serrated colorectal neoplasia. The mutational landscape of WNT signaling regulators is a mosaic that is underpinned by mutations in key driver genes, such as *RNF43* and *APC*. The mutations of *RNF43* and *APC* are mutually exclusive. We identified potential cancer driver genes in the WNT signaling axis. *MEN1* has previously been implicated in cancers of endocrine origin, but has not been identified as a tumor suppressor gene in colorectal cancer. We have identified a hotspot mutation in *MEN1* that effects 4% of *BRAF* mutant cancers. We have identified WNT16 as a potential driver gene by mutational hotspot analysis. WNT16 is a competitive inhibitor of canonical WNT and the mutation of WNT16 is common in *BRAF* mutant cancers. *BRAF* mutant colorectal cancers with truncating *APC* mutation tended to arise earlier in life, and presented at a significantly later stage. These cancers are extremely aggressive and the survival of patients with both *BRAF* and *APC* mutation is poor (12% 5-year survival). The in vivo modelling of *Apc* and *Braf* mutation revealed a dramatically increased tumor burden, with a median survival of 3.2 months for animals with both mutations. Therefore, we conclude that the co-mutation of *BRAF* and *APC* in colorectal cancers is conducive to an aggressive phenotype.

## Figures and Tables

**Figure 1 cancers-12-01171-f001:**
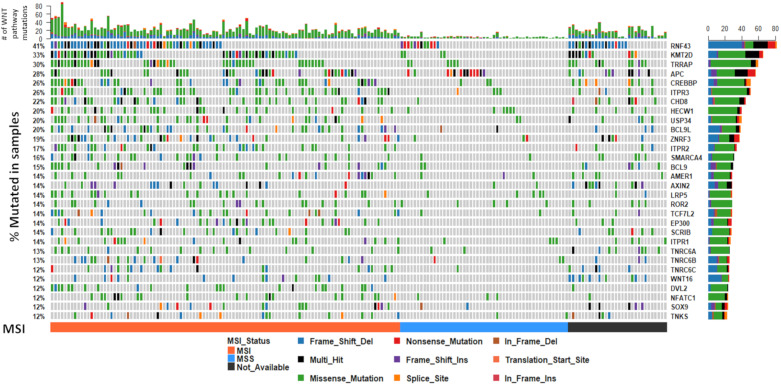
The somatic mutation landscape of WNT signaling regulators in BRAF mutant colorectal cancers. The 30 most frequently mutated genes in the WNT pathway are depicted. Each column corresponds to a single cancer. The colors of bars are indicative of the type of mutation, with grey = wild-type. The barplot at the top of the figure represents the number of mutations in the WNT pathway a sample has. The vertical plot on the right of the figure represents the number of mutations in each gene, color coded by mutation type. Microsatellite instability status is indicated below the mutation plot.

**Figure 2 cancers-12-01171-f002:**
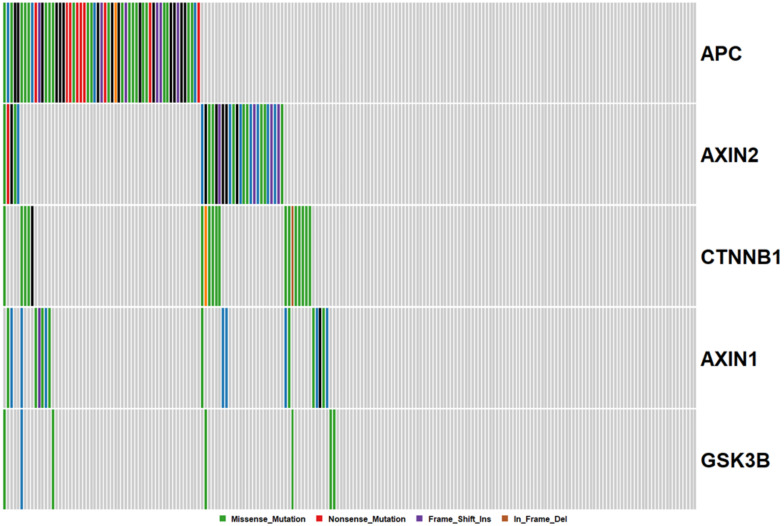
Mutations in the Beta-Catenin destruction complex. Each column corresponds to a single cancer, and each row, a single gene.

**Figure 3 cancers-12-01171-f003:**
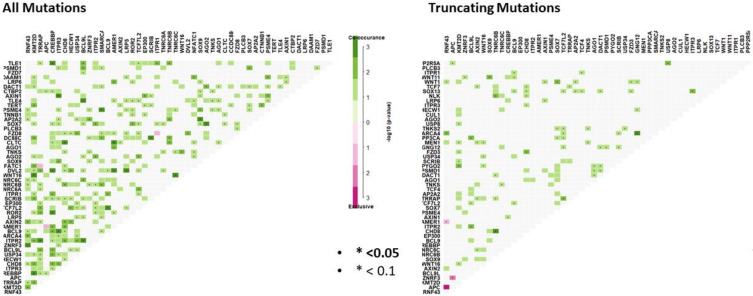
Somatic interaction analysis reveals mutually exclusive mutations between gene pairs, and significant co-occurring mutations. Co-occurring mutations are indicated by green squares and mutually exclusive mutations between gene pairs in purple. The intensity of the color is proportionate the –log10 (*p*-value). *p*-values were determined using Fisher’s exact test.

**Figure 4 cancers-12-01171-f004:**
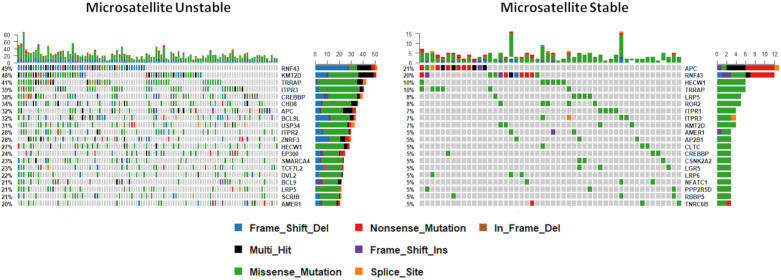
The mutational landscape of WNT signaling regulators by microsatellite instability status. Note that this analysis is limited to cancers with microsatellite instability status available (*n* = 167).

**Figure 5 cancers-12-01171-f005:**
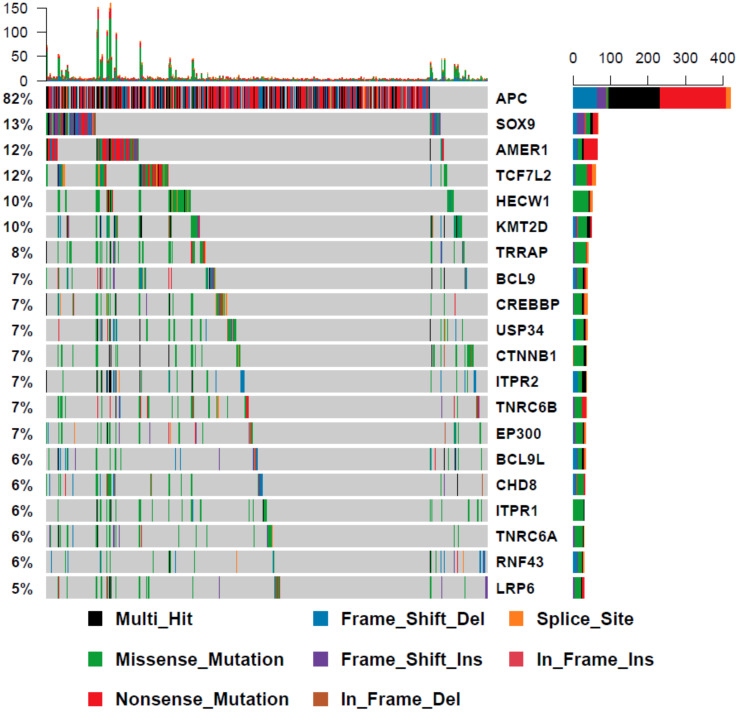
The mutational landscape of WNT signaling regulators in BRAF wild type cancers.

**Figure 6 cancers-12-01171-f006:**
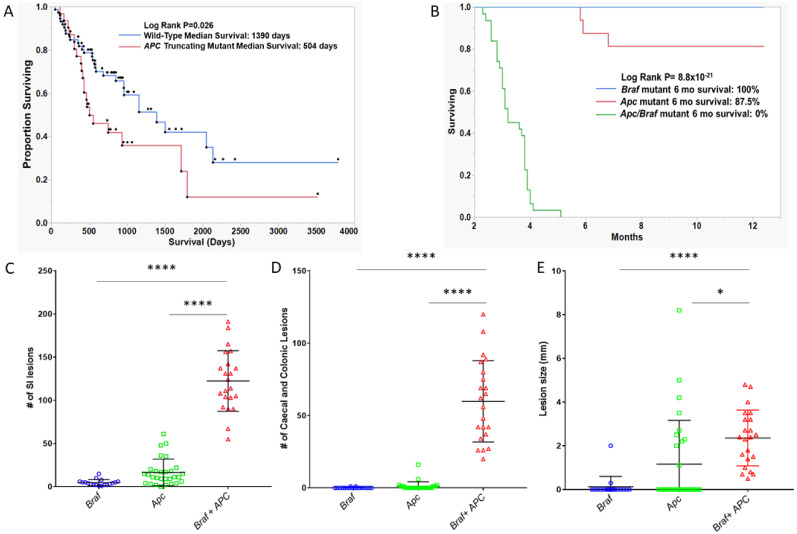
Survival analysis of (**A**) *BRAF* mutant human cancers by the presence or absence of truncating *APC* mutation. (**B**) *Apc*, *Braf*, and *Apc/Braf* mutant *murine* models. *p*-values are univariate and derived from the log-rank test. (**C**–**E**) Assessment of the number and size of lesions in *Apc*, *Braf*, and *Apc/Braf* mutant mouse models. (**C**) Total lesions in the small intestine. (**D**) Total number of lesions in the colon and caecum. E: Mean size of lesions in the colon and caecum.

**Table 1 cancers-12-01171-t001:** WNT signaling regulators that are differentially mutated according to microsatellite instability status in BRAF mutant colorectal cancers.

Gene	Samples Mutated (MSI)	Samples Mutated (MSS)	Adjusted *p* Value	Odds Ratio (MSI/MSS)
*KMT2D*	51	4	1.15 × 10^−06^	13.03657408
*CHD8*	36	1	1.15 × 10^−05^	30.45914642
*ITPR2*	30	0	1.68 × 10^−05^	NA
*CREBBP*	40	3	3.65 × 10^−05^	11.57936393
*ITPR3*	41	4	9.91 × 10^−05^	8.88801381
*BCL9L*	34	2	9.91 × 10^−05^	13.77175126
*EP300*	25	0	1.32 × 10^−04^	NA
*USP34*	33	2	1.46 × 10^−04^	13.18635308
*DVL2*	23	0	2.29 × 10^−04^	NA
*TRRAP*	43	6	2.86 × 10^−04^	6.194964561
*SCRIB*	22	0	3.48 × 10^−04^	NA
*RNF43*	52	12	2.09 × 10^−03^	3.900624188
*FZD8*	19	0	2.24 × 10^−03^	Inf
*ZNRF3*	30	3	2.34 × 10^−03^	7.558622414

**Table 2 cancers-12-01171-t002:** Comparison of the mutational frequency of WNT signaling regulators in BRAF mutant cancers versus BRAF wild type cancers.

Gene	Samples Mutated (BRAF Mutant)	Samples Mutated (BRAF Wild Type)	Adjusted *p* Value	Odds-Ratio (Mutant/Wild Type)
*APC*	57	420	7.18 × 10^−39^	0.09
*RNF43*	82	29	2.90 × 10^−26^	11.52
*ZNRF3*	38	7	3.67 × 10^−14^	16.84
*ITPR3*	51	25	3.27 × 10^−12^	6.65
*KMT2D*	66	49	2.53 × 10^−11^	4.64
*TRRAP*	60	40	2.53 × 10^−11^	5.04
*CREBBP*	51	37	9.61 × 10^−09^	4.38
*DVL2*	24	6	6.25 × 10^−08^	11.45
*FZD8*	20	3	8.60 × 10^−08^	18.77
*CHD8*	45	32	8.92 × 10^−08^	4.34
*TERT*	19	3	2.34 × 10^−07^	17.74
*LRP5*	29	13	2.72 × 10^−07^	6.49
*WNT16*	25	9	3.74 × 10^−07^	7.96
*SCRIB*	28	16	1.10 × 10^−05^	5.03
*ROR2*	29	18	1.10 × 10^−05^	4.64
*BCL9L*	39	33	1.76 × 10^−05^	3.51
*AKT1*	16	4	1.95 × 10^−05^	11.00
*SOX3*	17	5	1.99 × 10^−05^	9.38
*WNT4*	13	2	3.38 × 10^−05^	17.66

**Table 3 cancers-12-01171-t003:** Univariate survival analysis of the most frequently mutated WNT signaling regulators, and WNT signaling regulators identified as potential drivers by computational analysis. Differences in overall survival were assessed using the log-rank test.

Gene	Median Survival (Days)					
	All Mutations			Truncating Mutations		
	Wild Type	Mutant	*p* Value	Wild Type	Mutant	*p* Value
*WNT16 ^*	2134	547	0.001	2134	547	0.001
*RNF43 **	752	2047	0.01	934	2047	0.04
*CREBBP **	958	2134	0.01	961	NA	0.14
*APC **	1390	504	0.03	1390	504	0.03
*WNT11 ^*	2134	188	0.059	2134	188	0.059
*AXIN2 **	958	1158	0.26	961	1158	0.21
*KMT2D **	958	1503	0.27	961	1503	0.4
*AXIN1 **	958	1503	0.35	961	1158	0.86
*SOX9 **	961	2047	0.49	1158	2047	0.88
*GNG12 ^*	2134	NA	0.52	2134	1503	0.93
*MEN1 ^*	NA	2134	0.54	NA	2134	0.54
*PYGO2 ^*	NA	1818	0.63	NA	1818	0.5
*ZNRF3 ^*	2134	2047	0.65	2134	2047	0.9
*TRRAP ^*	NA	2047	0.8	2134	NA	0.81

* *n* = 109, ^ *n* = 50, NA = Indeterminable.

**Table 4 cancers-12-01171-t004:** Clinical and molecular characteristics of *BRAF* mutant cancers with *APC* mutation.

			*APC*			*p*-Value ^1^		
Characteristic		*n*	Truncating Mutation	Missense Mutation	Wild-Type	Truncating vs. Missense	Truncating vs. Wild-Type	Missense vs. Wild-Type
Mean Age	Years	273	60.8	72.4	70.6	**2.03 × 10^−5^**	**9.3 × 10^−6^**	0.34
Sex	Male	87 (31.8%)	18 (32%)	6 (24%)	63 (33%)	0.48	0.86	0.36
	Female	187 (68.2%)	39 (68%)	19 (76%)	129 (67%)			
Tumour Side	Left	42 (16.6%)	13 (25%)	1 (4%)	28 (16%)	**0.02**	0.16	0.08
	Right	211 (83.4%)	40 (75%)	23 (86%)	148 (84%)			
Stage	I	32 (12.9%)	3 (6%)	2 (9%)	27 (16%)	**0.01**	**0.002**	0.32
	II	93 (37.7%)	14 (27%)	12 (55%)	67 (39%)			
	III	59 (23.9%)	10 (19%)	6 (27%)	43 (25%)			
	IV	63 (25.5%)	25 (48%)	2 (9%)	36 (21%)			
CIMP	High	128 (81.0%)	20 (83%)	18 (95%)	90 (78%)	0.23	0.57	0.05
	Negative	30 (19.0%)	4 (17%)	1 (5%)	25 (22%)			
MSI	MSI	136 (54.8%)	22 (42%)	21 (91%)	93 (54%)	**5.3 × 10^−5^**	0.14	**0.0002**
	MSS	112 (45.2%)	30 (58%)	2 (9%)	80 (46%)			

^1^*p*-values were obtained using the likelihood–ratio test for categorical variables, and the Student’s *t*-test for continuous variables. All statistical analyses were two-tailed. Bold text indicates significance at *p* < 0.05

**Table 5 cancers-12-01171-t005:** Cox-proportional Hazard analysis of survival of *BRAF* mutant cancers.

Variable	Risk Ratio	95% CI	*p* Value
Microsatellite Instability (MSS)	2.41	1.18–4.95	0.016
Gender (Female)	1.93	1.04–3.57	0.037
APC (Truncating)	1.63	0.80–3.32	0.17
Stage (III/IV)	1.56	0.66–3.69	0.31
Age (<50)	1.2	0.53–2.71	0.65

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
