# Peer review of "APC Mutation Marks an Aggressive Subtype of BRAF Mutant Colorectal Cancers"

_cancers, 2020, doi:10.3390/cancers12051171_

Round 1

Reviewer 1 Report

The authors responded/addressed all the concerns raised adequately.

Reviewer 2 Report

The authors have adequately addressed the issues I raised.

This manuscript is a resubmission of an earlier submission. The following is a list of the peer review reports and author responses from that submission.

Round 1

Reviewer 1 Report

In this manuscript the authors examined the prognostic significance of WNT pathway genomic alterations in BRAF mutant colorectal cancers (CRCs).  The results of this study revealed that WNT pathway effector genes are frequently mutated in BRAF mutant colorectal cancer. Co-mutation of BRAF and APC was associated with poor outcome.

This is an excellent study to stratify high risk CRC patients. The manuscript is easy to follow.

1- It is well known in the literature that BRAF mutant MSS colorectal cancers represent an aggressive phenotype with poor outcome. For practical purposes, it would be informative if the authors could comment on whether the combination of BRAF/APC mutation better stratifies high risk CRC patients than combination of BRAF/MSS. In other words, does BRAF/APC mutant CRCs with MSI-H are associated with poor outcome?

2- It is a bit of challenge as to how the results of this study (with respect to different types of APC mutations) could be used in clinic to stratify high risk patients. It seems that APC mutations in general infer a poor outcome in setting of BRAF mutant CRCs. While it was shown that mainly truncated type APC mutations are associated with aggressive phenotype, this type of mutation did not turn out to be significant in a multivariate analysis.

3- Was there any WNT pathway mutation other than APC (e.g. RNF43, MEN-1, WNT-16, etc) that could predict worse survival? If not, please mention this in the manuscript.

4- Please add the markers used for definition of CIMP in the material and method section and also report the number of cases that have CIMP information available.

5- As the authors mentioned in the manuscript, the mouse model used in this study is not the best model to represent BRAF/APC mutant CRC in human. It seems that the reduced survival in mice was related to the polyp burden rather than developing malignancy. On the other hand, the combination of BRAF and APC mutations in non-dyplastic/non-malignant human colonic polyps is rare. BRAF mutation reportedly leads to senescence and this model may represent an escape mechanism by acquiring APC mutation.

Author Response

We thank the reviewers for their valuable contributions and comments to our study and have added the following to the acknowledgements section of the manuscript:

“We acknowledge the contributions of the reviewers and their insightful comments that helped shape the manuscript.”

Reviewer 1

In this manuscript the authors examined the prognostic significance of WNT pathway genomic alterations in BRAF mutant colorectal cancers (CRCs).  The results of this study revealed that WNT pathway effector genes are frequently mutated in BRAF mutant colorectal cancer. Co-mutation of BRAF and APC was associated with poor outcome.

This is an excellent study to stratify high risk CRC patients. The manuscript is easy to follow.

  • It is well known in the literature that BRAF mutant MSS colorectal cancers represent an aggressive phenotype with poor outcome. For practical purposes, it would be informative if the authors could comment on whether the combination of BRAF/APC mutation better stratifies high risk CRC patients than combination of BRAF/MSS. In other words, does BRAF/APC mutant CRCs with MSI-H are associated with poor outcome?

This is an important point. In this study, we did not have enough BRAF/APC mutant MSI-H cancers with survival data (n=12) to determine whether APC mutation in this setting confers poorer survival outcomes. Interestingly, however, when we consider BRAF/APC mutant MSS cancers alone, we observe a borderline significant association (P=0.052) with poorer overall survival in BRAF mutant MSS cancers with truncating APC mutations. This suggests that APC mutation confers a survival disadvantage to these patients that is above what is conferred by microsatellite stability. We have added the following to the manuscript:

…This effect was most pronounced in microsatellite stable cancers (Supplementary figure S3A) compared with microsatellite unstable cancers (Supplementary Figure S3B & S3C).”

  • It is a bit of challenge as to how the results of this study (with respect to different types of APC mutations) could be used in clinic to stratify high risk patients. It seems that APC mutations in general infer a poor outcome in setting of BRAF mutant CRCs. While it was shown that mainly truncated type APC mutations are associated with aggressive phenotype, this type of mutation did not turn out to be significant in a multivariate analysis.

In this study we show that truncating mutations in APC are associated with poorer overall survival. As these mutations were also associated with microsatellite stable (MSS) cancers, we performed multivariate analyses to determine whether the survival disadvantage was independent of microsatellite instability. We did not find strong evidence that APC mutation was independently associated with survival (P=0.17), however we identified a borderline significant association between poorer survival in microsatellite stable cancers with truncating APC mutations. As BRAF mutant MSS cancers already have dismal prognoses, our study have identified a subgroup that do even more poorly.

  • Was there any WNT pathway mutation other than APC (e.g. RNF43, MEN-1, WNT-16, etc) that could predict worse survival? If not, please mention this in the manuscript.

We have now performed survival analysis on the top 5 most frequently mutated genes and genes identified as potential drivers by oncodriveCLUST and oncodriveFML. We have reported any non-silent mutation versus wild type, and truncating mutations vs non-truncating mutations and wild type for each gene. We have provided the p-values for these analysis in tabular format.

We have added the following results paragraph:

“Mutations in WNT signaling regulators confer prognostic implications

To determine whether frequently mutated WNT regulators, or those identified as putative cancer drivers is relevant to patient outcome we performed survival analysis on patients with cancers bearing mutations in the five most frequently mutated WNT signaling regulators (RNF43, KMT2D, TRRAP, APC, CREBBP) and those identified by driver analysis (MEN1, GNG12, WNT16, AXIN1, AXIN2, ZNRF3, SOX9, BCL9L, PYGO2 and WNT11).  Four genes (APC, RNF43, CREBBP and WNT16) were associated with survival (Table 1). Mutations in WNT16 and APC were significantly associated with shorter overall survival (Table 1). By contrast, mutations of CREBBP and RNF43 were associated with prolonged survival (Table 1).”

We are conscious that these analyses may have been underpowered, especially for low frequency mutations in genes such as MEN1 and GNG12. We have added the following to the discussion describing the limitations of these analyses:

“Although our study is the largest to date to assess WNT regulators in BRAF mutant cancers, we did not have a sufficiently large enough sample size to draw conclusions as to the prognostic implications of genes that are mutated more rarely, such as MEN1 and GNG12. Mutations in hotspots, or in relatively small genes could be assessed in much larger cohorts using less expensive genotyping technologies to determine whether mutations in these genes are relevant to patient outcome.”

  • Please add the markers used for definition of CIMP in the material and method section and also report the number of cases that have CIMP information available.

The method of assessing CIMP was not uniform across studies. For example, the TCGA determined CIMP by microarray based clustering, whereas other studies used standard marker panels. As CIMP was not considered a significant factor in this study, we do not believe this limitation alters the findings or conclusions we have drawn. We have added the following text to the discussion section of the manuscript acknowledging as much:

“…Our study was also limited in the ability to assess the role of the CpG island methylator phenotype in WNT signaling. CIMP was assessed in some samples in our study cohorts, however the method of assessment was not uniform across samples. For example, cancers from TCGA assessed CIMP via microarray based clustering, in contrast other studies used qPCR based marker panels. Moreover many WNT signaling genes are putative methylation targets. Further study is necessary to assess the role of CIMP in shaping WNT signaling in the context of BRAF mutant colorectal cancer.”

5- As the authors mentioned in the manuscript, the mouse model used in this study is not the best model to represent BRAF/APC mutant CRC in human. It seems that the reduced survival in mice was related to the polyp burden rather than developing malignancy. On the other hand, the combination of BRAF and APC mutations in non-dyplastic/non-malignant human colonic polyps is rare. BRAF mutation reportedly leads to senescence and this model may represent an escape mechanism by acquiring APC mutation.

The reviewer raises an interesting point. BRAF mutation has been reported to induce senescence, which can be overcome through silencing of the CDKN2A loci. It has been previously reported that APC loss in the intestine does require loss of p21 to generate adenomas and does not induce senescence. It is possible that APC mutations, in the context of BRAF mutation, facilitate an exit from oncogene induced senescence thereby facilitating malignant transformation.

We have added this as a point of discussion in the manuscript:

“…BRAF mutation has been reported to induce senescence, which can be overcome through silencing of the CDKN2A loci50. It has been previously reported that APC loss in the intestine does require loss of p21 to generate adenomas and does not induce senescence51. It is possible that APC mutations, in the context of BRAF mutation, facilitate an exit from oncogene induced senescence and while simultaneously satisfying the requirement for WNT signaling, facilitating a rapid transition to dysplasia and invasive cancer.”

Reviewer 2 Report

The authors analyzed the whole exome sequencing data of 199 BRAF mutant colorectal cancer samples, incorporating clinical data such as tumor aggressiveness, tumor volume, age, gender, etc.  They assessed the alterations on WNT pathway and their relation such as mutual-exclusivity and co-occurrence. They identified APC-truncating mutations and RNF43 mutations to be mutually exclusive. They also identified potentially novel drivers in WNT pathway, such as WNT16. Correlation of the clinical data with the mutation data revealed that BRAF-APC co-mutant cases displayed a more aggressive presentation.

Major comments:

  1. WNT16 and MEN1 are presented as the major findings of the study as they are potentially novel drivers. However, the focus of the paper seems to be on the clinical correlations of the APC/BRAF mutant groups. Details of the analysis that identified especially WNT16 should be presented. Moreover, clinical correlation similar to the BRAF/APC mutations should be done, and must be mentioned even if reveals to be non-significant.
  2. Even though the mutation burden of the samples is presented as a supplementary figure, mismatch repair deficient status of the samples and the relation to the findings should be discussed/analyzed more.
  3. Mutual exclusivity of APC and RNF43 has been previously reported in PMID: 25344691. This paper is already cited but in another context. The fact that the finding presented was previously reported should be highlighted and why the findings in the presented study is different/important should be better explained.
  4. In methods section, it is not clear whether both the normal and cancer tissue were sequenced to the same depth
  5. Somatic mutation calling algorithms are not recent methods. Haplotype caller is no longer advised for somatic calling—per GATK best practices. More up-to-date mutation calling algorithms should be used such as mutect/mutect2
  6. Details of the oncodriveclust and oncodriveFML methods should be given--the parameters etc. should be reported to assure reproducibility
  7. The reasoning why the non-WNT mutations were excluded should be explained. If data is fed into non-biased driver identification methods, results might significantly be affected, as some of these tumors are MSI/hypermutated. By excluding non WNT mutations, a bias is created and potential discovery of other drivers is hampered. The analysis should be repeated with all mutation data set.
  8. In Figure 1, total mutation burden/MSI status should also be depicted as it also has implications on clinical outcome
  9. Clinical outcome, tumor aggressiveness is highly correlated with MSI status. The findings should be analyzed/discussed more in relation to MSI status, it should be corrected for that to show dependency/independency.
  10. Are there any APC double hits in BRAF mutant cases?
  11. VAF comparison of BRAF/APC and other WNT alterations is not mentioned. This data should be and a detailed analysis should be performed to assess the clonality and timing of these co-occurring and mutually exclusive mutations.
  12. Conclusions regarding WNT16 very hypothetical conclusion-no functional or transcriptomic data to prove the claims is provided

Minor comments:

  1. Section 2.3, line 6—typo “As we sought to investigator …”
  2. Supplementary S1.xlsx-sheet 2 does not have column names, the table is not interpretable.

Reviewer 3 Report

Fennell et al. evaluated the mutational landscape of WNT signaling regulators in colorectal cancer (CRC) with BRAF mutation using 24 in-house exome data and 175 publicly available exome data. The authors demonstrated frequent mutation of genes in WNT signaling in BRAF-mutated CRC. The authors also demonstrated that APCMin/+/BRAFV637 mice showed more aggressive phenotype compared to APCMin/+ mice or BRAFV637 mice. The study is potentially interesting, but some major issues should be addressed.

(Major points)

  1. The authors should include quality assessment of their own exome data to see if the data can be combined with publicly available exome data.
  2. How mutational profiles of WNT signaling is different between BRAF-wild and mutant CRC?
  3. The authors should use OncodriveCLUSTL instead of OncodriveCLUST.
  4. The authors should discuss possible mechanisms associated with APC mutation-induced aggressiveness in the context of BRAF mutant CRC.

Author Response

Reviewer 3

Fennell et al. evaluated the mutational landscape of WNT signaling regulators in colorectal cancer (CRC) with BRAF mutation using 24 in-house exome data and 175 publicly available exome data. The authors demonstrated frequent mutation of genes in WNT signaling in BRAF-mutated CRC. The authors also demonstrated that APCMin/+/BRAFV637 mice showed more aggressive phenotype compared to APCMin/+ mice or BRAFV637 mice. The study is potentially interesting, but some major issues should be addressed.

(Major points)

  1. The authors should include quality assessment of their own exome data to see if the data can be combined with publicly available exome data.

We have now included supplementary data outlining sequencing quality control metrics. Please see supplementary table S3 for further details.

  1. How mutational profiles of WNT signaling is different between BRAF-wild and mutant CRC?

We have now assessed the mutational profiles of 512 BRAF wild type cancers from the TCGA. We have added one figure (Figure 5) and one table (Table 2) describing these results. The following has been added to the results

“WNT signalling regulators are differentially mutated in BRAF mutant cancers in comparison to BRAF wild type cancers

To establish whether the WNT mutational landscape of colorectal cancers is differs according to BRAF status we examined the mutational profile of WNT signalling genes in further 512 colorectal cancers from The Cancer Genome Atlas that were BRAF wild type. The most frequently mutated genes are markedly different in BRAF wild type cancers (Figure 5, Figure 1). APC was mutated in 82% of these cancers, in comparison to just 28% of BRAF mutant cancers. Of all WNT signalling regulators assessed in this study, 110 were significantly differentially mutated between BRAF mutant and wild type cancers (Table 2). 99% of these genes were more frequently mutated in BRAF mutant cancers. This is likely due to the increased frequency of MSI in BRAF mutant cancers, and the high frequency of the mutations in highly penetrant APC gene in BRAF wild type cancers reducing the selective pressure on other WNT regulators.”

  1. The authors should use OncodriveCLUSTL instead of OncodriveCLUST.

As new algorithms are continually developed it is inevitable that new tools may become available that could deliver a different insight into the data analysis but in order to produce a finalized analysis the bioinformatics tools and analysis at some stage must be selected and completed.  OncodriveCLUSTL is a new algorithm published at the end of last year that reports superiority over its previous incarnation OncodriveCLUST, that we use here, however, the newer implementation excludes the analysis of insertions and deletions (indels, see below from the documentaton). We feel that the results from OncodriveCLUST that includes indels is important and relevant as many of these cancers are microsatellite unstable, therefore we anticipated indel hotspots and sought to detect these.

“Mutations file. TSV file containing SNVs (substitutions) mapped to a reference genome (e.g., human hg19 or mouse c3h). If other mutation types are present (insertions, deletions, double base substitutions, etc), they will be filtered out during the analysis. This file must contain, at least, the following 5 columns with header:” https://bitbucket.org/bbglab/oncodriveclustl/src/master/

We attempted to run OncodriveCLUSTL, however, as expected, the newer algorithm did not detect the hotspot indel mutations in WNT16. Further, The hotspot indel mutations reported by OncodriveCLUST include MEN1 and GNG12 were not detected by OncodriveCLUSL, but were also detected by OncodriveFML. 

  1. The authors should discuss possible mechanisms associated with APC mutation-induced aggressiveness in the context of BRAF mutant CRC.

We have added the following text discussing how APC mutation may promote the rapid progression of BRAF mutant precancerous lesions:

The mechanisms associated with APC mutation induced aggressiveness in the context of BRAF mutant cancers are not clear. Aberrant WNT signalling appears to be a prerequisite for the acquisition of dysplasia in traditional serrated adenomas and sessile serrated lesions. In traditional serrated adenomas, which are by nature dysplastic, are thought to have transitioned from hyperplastic polyps or sessile serrated lesions, and this transition coincides with the acquisition of WNT pathway alterations, such as RSPO fusions and RNF43 mutations. In sessile serrated lesions that acquire dysplasia, RNF43 mutations are extremely common (>50%), however are rare in lesions that are not yet dysplastic. This is an important distinction, as lesions with dysplasia are thought to progress within 12 months to invasive cancer. Alterations to RNF43 are dependent on DNA methylation induced loss of MLH1 and mismatch repair function. DNA methylation may develop over a protracted period of time, increasing the sojourn of these lesions to dysplasia and ultimately malignancy. Genetic acquisition of an APC mutation, when it occurs, is likely to immediately satisfy the requirement for WNT signalling and facilitate a rapid transition to dysplasia and invasive cancer. This may explain why precancerous lesions with both APC mutation and BRAF mutation are rarely identified, and why the cancers with these mutations present at a younger age.”